# Parasitic Parameters Extraction of High-Speed Vertical-Cavity Surface-Emitting Lasers

Haixia Tong [1] , Yanjing Wang [2], Sicong Tian [2], Cunzhu Tong [2,3,*], Zhipeng Wei [1,*] and Lijun Wang [2]

1   State Key Laboratory of High Power Semiconductor Lasers, School of Physics, Changchun University of Science and Technology, Changchun 130022, China; tonghaixia18@mails.ucas.ac.cn
2   State Key Laboratory of Luminescence and Applications, Changchun Institute of Optics, Fine Mechanics and Physics, Chinese Academy of Sciences, Changchun 130033, China; wangyanjing@ciomp.ac.cn (Y.W.); tiansicong@ciomp.ac.cn (S.T.); wanglj@ciomp.ac.cn (L.W.)
3   Jlight Semiconductor Technology Co., Ltd., Changchun 130033, China
*   Correspondence: tongcz@ciomp.ac.cn (C.T.); weizp@cust.edu.cn (Z.W.); Tel.: +86-0431-86176349 (C.T.); +86-0431-85583390 (Z.W.)

**Abstract:** Parasitic parameters, including electrical capacity and inductance, are the key limiting factors for bandwidth improvement of high-speed vertical-cavity surface-emitting lasers (VCSELs). The traditional parasitic extraction method, which uses a first-order low-pass filter transfer function, is oversimplified, and there are large deviations between the obtained data and the actual measured data. In this paper, we proposed a modified parasitic extraction method that described the extrinsic behaviour of the high-speed oxide-confined VCSELs well and easily extracted the values of all parasitic parameters. This method can also precisely fit microwave reflection coefficient $S_{11}$ data even at high frequencies and provide design guidance for high-speed VCSELs. Using this method, a high-speed 850 nm VCSEL featuring a six-layer oxide aperture with −3 dB bandwidth up to 23.3 GHz was analysed. The electrical parasitics have been systematically extracted from VCSELs with different oxide apertures. The enhanced bandwidth based on the improvement of parasitic parameters was discussed. It was found that the critical parasitic factors that affect the −3 dB bandwidth of VCSELs are pad capacitance and inductance.

**Keywords:** high-speed VCSEL; parasitic extraction method; impedance characteristics

## 1. Introduction

VCSELs have important applications in short-range optical interconnects due to their low cost, low threshold, low power consumption, and high speed. In recent years, the development of big data, cloud services, and supercomputers has increased the demand for short-range optical interconnects [1] and higher speed and bandwidth are expected for 850 nm VCSEL technology. To realize higher bandwidth, it is essential to deeply understand the intrinsic dynamics and bandwidth limiting effects of VCSELs. High bandwidths of 28 GHz and 24 GHz were achieved by using four AlGaAs oxide layers with a low aluminium component and two primary oxide layers and four secondary oxide layers on the p-side of the active areas [2,3], respectively. An 850 nm VCSEL with a bandwidth of 28.2 GHz was demonstrated using InGaAs/AlGaAs quantum wells and four oxide layers [4]. The recorded bandwidth of the VCSEL was 30 GHz [5]. A commonly optimized parasitic response method was used to analyse device parasitism by extracting parasitic parameters. Parasitics can be isolated from the reflection coefficient by using numerical optimization techniques [6–14]. The simplest circuit model is a first-order low-pass filter containing only $C_a$, $R_a$, and $R_m$. As the authors of [14] mentioned, the approach of extracting data using a first-order low-pass filter transfer function is oversimplified, and there are large deviations between the obtained data and the actual measured data. Nevertheless, in most of the literature [15–17], this approach is still accepted as an approximation to the

exact extrinsic transfer function. In addition, the equations of the plate capacitor presented in [14,18] are inappropriate for extracting the pad capacities and resistance of the device.

In this paper, we proposed a modified parasitic extraction method that extracts the parasitic data directly by using the least squares approach based on the equivalent circuit model. The method described the extrinsic behaviour of the high-speed oxide-confined VCSELs well and easily extracted the values of all parasitic parameters. In addition, we designed, fabricated, and characterized 850 nm VCSELs with six oxide-confined layers, and the parasitic values of the devices with different diameters of oxide apertures and mesa have been systematically compared and analysed. Finally, suggested improvements are discussed to improve the parasitic bandwidth and address the performance enhancement limitations.

## 2. Materials and Methods

### 2.1. Fabrication and Performance of the Six-Layer Oxide Aperture VCSEL

There are many ways to improve the bandwidth of the device, and we focused mainly on the improvement of external parasitics to obtain the maximum small-signal modulation bandwidth. The epitaxial structure was grown by MOCVD, which consists of four secondary oxide layers above the upper primary oxide aperture, to reduce the capacitance. Four InGaAs quantum wells with five AlGaAs barriers were used as the active medium in the $\lambda/2$ cavity VCSEL. The distributed Bragg reflectors (DBRs) were optimized with graded interfaces [5]. The n-doped bottom-DBR comprised $Al_{0.88}Ga_{0.12}As/Al_{0.12}Ga_{0.88}As$ layers and $AlAs/Al_{0.12}Ga_{0.88}As$ layers; AlAs layers can improve the dissipation of heat due to their high thermal conductivity [4]. For device fabrication, the wafer was processed into the first cylindrical mesas of 12–18 μm in diameter and etched down to the active region in $SiCl_4/Ar$ plasma by using an inductively coupled plasma etcher. Then, the second cylindrical mesas of 27–33 μm in diameter were etched down to the n-GaAs contact layer. The oxidation aperture was formed by wet oxidation with a diameter of 3–9 μm. Benzocyclobutene (BCB) was coated on the device. Figure 1a shows the top view of the VCSEL with ground-signal-ground (GSG) Ti/Au probe pads, and Figure 1b shows the scanning electron microscope image of the cross-section of the VCSEL.

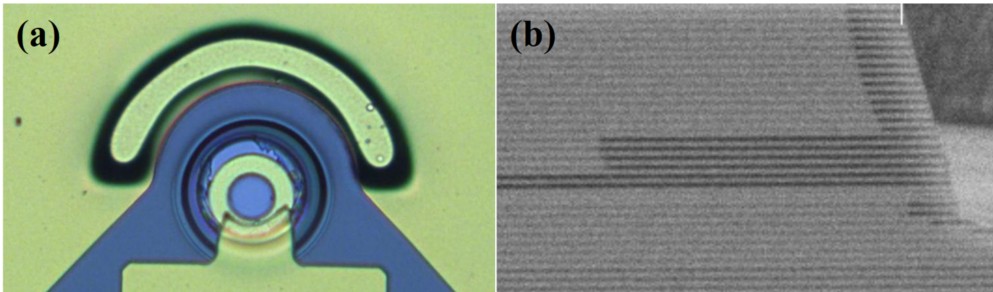

**Figure 1.** (**a**) Top view of the fabricated VCSEL with GSG probe pads, and (**b**) SEM image of the cross-section of the device after wet oxidation. The dark lines are the oxidized high-Al-composition layers.

Figure 2a shows the measured L-I-V curves and the lasing spectra of the fabricated VCSEL. The threshold current is 0.6 mA for a 3 μm oxide aperture, and the differential resistance is 70 Ω at 8 mA. The slope efficiency is approximately 0.45 W/A, and the maximum emission power exceeded 4 mW. Figure 2b shows the lasing spectra, which show that the device operated at a multimode of approximately 850 nm.

Figure 3a shows the small-signal modulation response of the VCSEL measured at 25 °C by a vector network analyser N5247B, and Figure 3b shows the resonance frequency and −3 dB bandwidth plotted against the square root of the bias current above the threshold for the analysis of the D-factor and modulation current efficiency factor (MCEF). The D-factor reflects the rate of increase of the relaxation frequency with current and determines the

modulation speed of an ideal laser [19]. It can be seen that the bandwidth reached 23.3 GHz at 12 mA, the MCEF is 9.04 GHz/mA$^{1/2}$, and the D-factor is 6.38 GHz/mA$^{1/2}$.

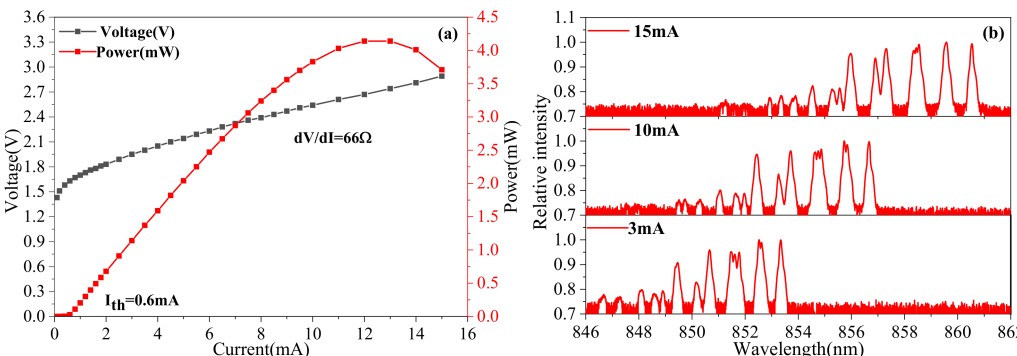

**Figure 2.** (**a**) L-I-V curves of the VCSEL with a 3 μm oxide aperture, and (**b**) lasing spectra at different bias currents.

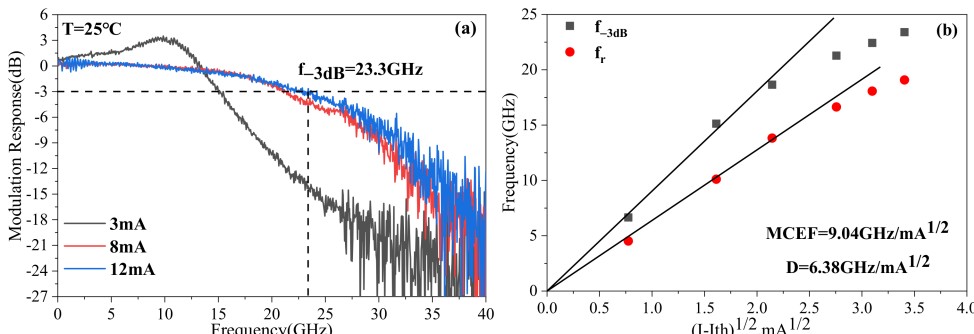

**Figure 3.** (**a**) Measured small-signal modulation response at different bias currents for the 3 μm oxide aperture VCSEL, and (**b**) the resonance frequency plotted against the square root of the bias current above the threshold with the D-factor and MCEF fits.

### 2.2. Equivalent Circuit Model

The modulation bandwidth can be improved by both changing the internal factors and minimizing the external parasitic components [20]. The parasitic parameters are structure dependent and can be extracted using an equivalent circuit model [19]. Figure 4 shows a schematic diagram of the oxide-confined VCSEL marked with the equivalent circuit, which includes all the major extrinsic parasitic effects. $L$ is the inductance, $C_p$ represents the frequency-dependent pad capacitance between the metal contact on the BCB resin and the bottom mirror stack at the ground, $R_p$ represents the frequency-dependent dielectric losses in the BCB, $R_{mirror}$ represents the series equivalent resistance of both the p-type and n-type DBR mirrors, $R_{sheet}$ is the sheet resistance along with the bottom contact, and $R_{contact}$ is the resistance of both p-type and n-type contact pads. The total resistance $R_m$ can be written as $R_m = R_{mirror} + R_{sheet} + R_{contact}$. $R_a$ is the resistance of the active region and is associated with the oxide aperture. $C_{oxide}$, $C_j$, and $C_{intrinsic}$ represent the capacitance of the oxide layer, junction, and intrinsic layer, respectively. The junction capacitance $C_j$ is the sum of the depletion capacitance and diffusion capacitance; however, under normal forward bias conditions, $C_j$ is dominated by the diffusion capacitance. $C_{intrinsic}$, $C_{oxide}$, and $C_j$ are grouped together as [21]: $C_a=C_j+(1/C_{oxide}+1/C_{intrinsic})^{-1}$.

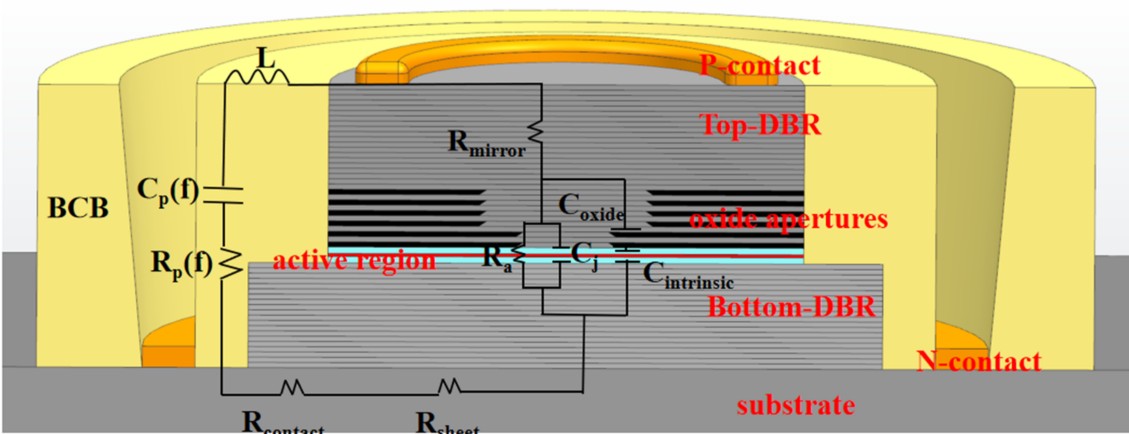

**Figure 4.** Equivalent electrical circuit model of an oxide-confined top-emitting VCSEL.

The values of the electrical components in the equivalent circuit model were obtained by fitting the measured reflection coefficient $S_{11}(f)$, which can be expressed as [15,22]:

$$S_{11}(f) = \frac{Z_T(f) - Z_0}{Z_T(f) + Z_0}, \qquad (1)$$

where $Z_T(f)$ is the total impedance and $Z_0$ denotes the characteristic impedance of the RF-driving source, which typically holds a value of 50 Ω, and

$$Z_T(f) = \left( \frac{1}{Z_{SM} + i2\pi f L} + \left( R_P + \frac{1}{i2\pi f C_p} \right)^{-1} \right)^{-1}, \qquad (2)$$

Here, $Z_{SM}$ represents the sub-input impedance containing the three circuit elements $R_m$, $R_a$ and $C_a$ and can be described as:

$$Z_{SM}(f) = R_m + \left( \frac{1}{R_a} + i2\pi f C_a \right)^{-1}, \qquad (3)$$

According to the extrinsic transfer function [14],

$$H_{ext}(f) = \frac{Z_0 i_a}{V_s} = \frac{Z_1(f)}{Z_2(f)} \cdot \frac{Z_4(f)}{Z_4(f) + Z_0} \cdot \frac{Z_0}{R_a}, \qquad (4)$$

where the impedance $Z_1(f)$ to $Z_4(f)$ can be expressed as shown in (5)–(8):

$$Z_1(f) = \frac{R_a}{1 + i2\pi f R_a C_a}, \qquad (5)$$

$$Z_2 = Z_1(f) + R_m + i2fL, \qquad (6)$$

$$Z_3(f) = \frac{1}{i2\pi f C_p} + R_p, \qquad (7)$$

$$Z_4(f) = \left( \frac{1}{Z_2(f)} + \frac{1}{Z_3(f)} \right)^{-1}, \qquad (8)$$

where $i_a$ is the small-signal current across the active region, and $V_s$ represents the small-signal voltage generated by the source.

### 2.3. Parasitic Extraction Method

Although the theoretical model is the same equivalent circuit model [14,21], the extracted parameters were determined by the fit method. In [15–17], the approach of

extracting data using a first-order low-pass filter transfer function, which is the simplest circuit model that contains only $C_a$, $R_a$, and $R_m$, is oversimplified. In addition, in [14,18], the equation of $C_p$ and $R_p$ used in the device parasitic value extraction is based on [23]. In [23], capacitors corresponding to the probe pads of VCSELs were fabricated above conducting ground planes on undoped GaAs substrates [23], this approach has not taken into account the effect of the device on pad capacitation and resistance. In this work, we extract the parasitic data employing the nonlinear least squares method that is based on the equivalent circuit model, which allows for avoiding inappropriate formulations and extracting various parasitic parameters directly simultaneously. The basic least squares approach to solving curve fitting problems is given as [24]:

$$f(x) = a_1 \varphi_1(x) + a_2 \varphi_2(x) + \cdots + a_m \varphi_m(x) \tag{9}$$

where $a_m$ ($m = 1,2,3,4,5 \ldots m < n$) is the undetermined coefficient and $\varphi_m(x)$ is a group of selected linearly independent functions.

The values of $L$, $R_m$, $R_a$, $C_a$, $R_p$, and $C_p$ can be easily obtained through this approach. Owing to the relationship between the values, the fitting result is multiple groups of values. The most reasonable group of data is the one that simultaneously satisfies the $S_{11}$ curve fitting minimum error; all values obey the rules of physics and demands of the judgement requirement $|H_{ext}(f)|^2 / |H_{ext}(0)|^2 = 0.5$ [24].

## 3. Results and Discussion

### 3.1. Parameter Extraction and Analysis

Figure 5a shows the $S_{11}$ data fit of our fabricated device. As seen in the figure, the data fitting curve has a relatively excellent fit to the measurement curves, and the fitting error is still kept to three decimal places despite the noise in the measurement data at high frequency.

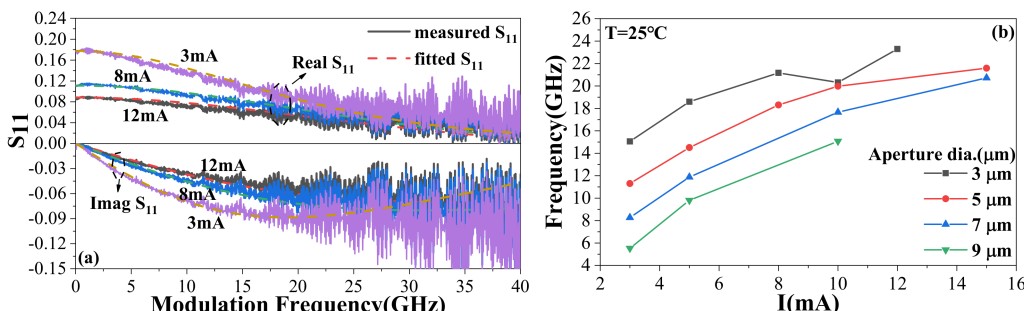

**Figure 5.** (**a**) Real and imaginary parts of the measured $S_{11}$ spectra of a 3 μm oxide aperture VCSEL. The measurement is performed at room temperature over a frequency range of 40 GHz at 3 mA, 8 mA, and 12 mA. Both parts are fitted using the calculated analytical function from the equivalent circuit model (dashed curve), and (**b**) the −3 dB frequency $f_{-3\,dB}$ of the 3, 5, 7, and 9 μm 850 nm oxide aperture VCSELs at different bias currents.

It is interesting to know how the −3 dB modulation bandwidth $f_{-3\,dB}$ and the values of the small-signal equivalent circuit elements change with the VCSEL oxide aperture diameter. Thus, we fabricated and utilized 3, 5, 7, and 9 μm oxide aperture diameter 850 nm VCSELs and extracted their parasitic data. Figure 5b shows the −3 dB bandwidth $f_{-3\,dB}$ of different oxide aperture VCSELs at different bias currents. As shown in the figure, the $f_{-3\,dB}$ of the same oxidation aperture device increases with the current in a logarithmic curve and decreases as the diameter of the oxide aperture increases.

Figure 6a–f show the polynomial fitting curve of extracted values and 95% fit confidence bounds of $R_p$, $C_p$, $C_a$, $R_a$, $R_m$, and $L$, respectively, as a function of different bias currents for the 3, 5, 7, and 9 μm 850 nm oxide aperture diameter VCSELs at 25 °C. From Figure 6a,b, we can observe that the values of $R_p$ and $C_p$ decrease with increasing cur-

rent. Additionally, the larger the mesa diameter, the smaller $R_p$ and $C_p$. According to Equation (4), $C_p \propto 1/f_{-3\,\text{dB}}$. The different amounts of variation in the $C_p$ value of the device at the same current are related to the different filling areas due to the different mesa diameters of the device and the different filling thicknesses during the device fabrication process. As shown in Figure 6c, the value of $C_a$ increases with increasing current and oxide aperture diameter. The oxide layer capacitance can be calculated by the plate capacitor formula [25], and the value of $C_a$ is directly proportional to the oxidized area.

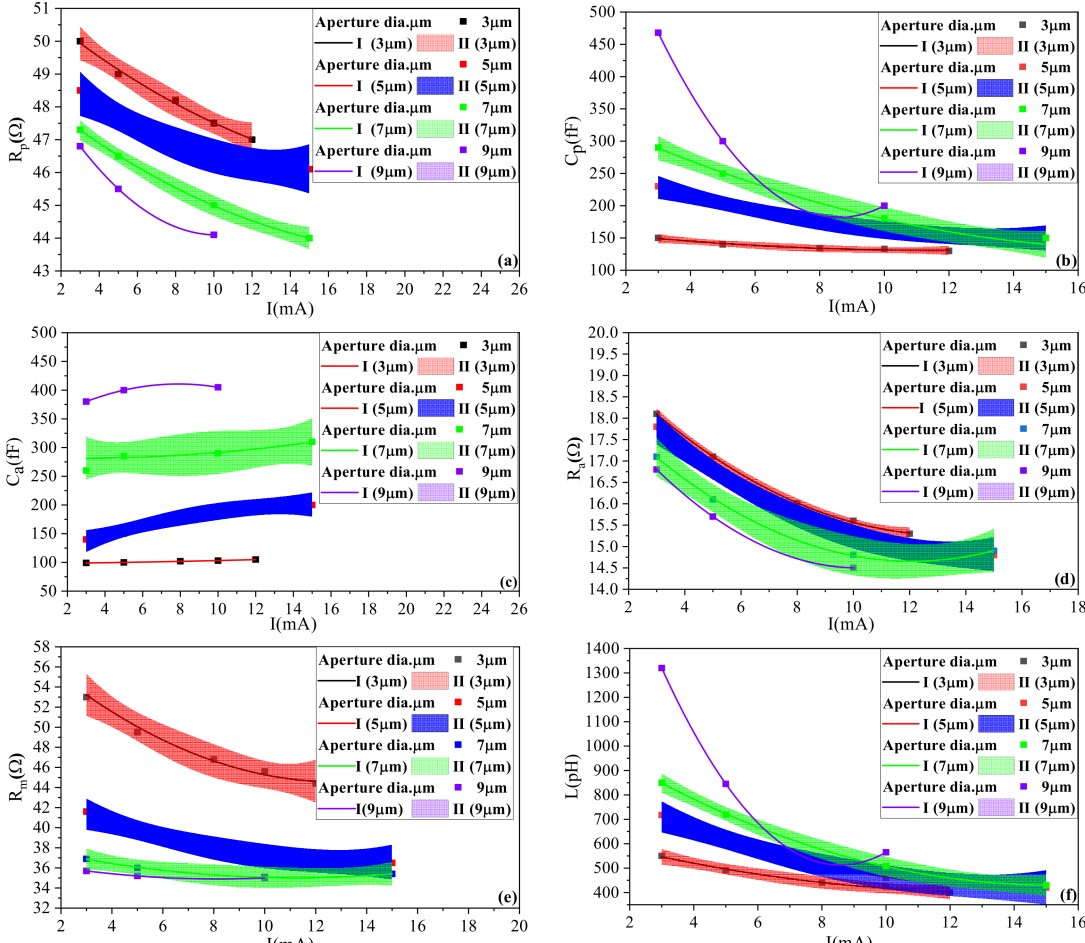

**Figure 6.** (**a**–**f**) show the extracted values and 95% fit confidence bounds of $R_p$, $C_p$, $C_a$, $R_a$, $R_m$, and $L$, respectively, as a function of different bias currents for the 3, 5, 7, and 9 μm 850 nm oxide aperture diameter VCSELs at 25 °C. In the legend, I is the polynomial fitting curve, II is the 95% fit confidence bounds.

As shown in Figure 6d,e, the resistance $R_a$, $R_m$ decreases with increasing current and mesa diameter. Figure 6f shows that the $L$ of the device decreases with increasing current and decreasing oxide aperture diameter. According to equation $L = (2\pi f \sigma \delta)^{-1}$, we know that $L \propto 1/f$. $\sigma$ is the electrical conductivity, $\delta$ is the skin depth, and $f$ is the frequency. From Equation (4), it is known that the device bandwidth increases as $L$ decreases. From Figure 6 can be seen that 95% fit confidence bounds are quite narrow, and the polynomial fitting curves are obeying the rules of physics, apart from the abnormal data of 9 μm oxidation aperture for $R_p$ and $C_p$ at a 3 mA current. The abnormal in individual data may be caused by test errors.

### 3.2. Bandwidth Improvement Suggestions

Since the 3 μm device has the highest bandwidth, we only analyse the effect of the parasitic value of the device on the bandwidth with a 3 μm oxide aperture size. Table 1

lists the $f_{-3\,\text{dB}}$ at different currents with a 30% reduction of $L$, $C_a$, $R_a$, $R_m$, $R_p$, and $C_p$ and compares the results with the original values.

**Table 1.** Dependence of the parasitic −3 dB frequency on 30% reductions of various circuit elements.

| Current (mA) | Original Value of $f_{-3\,\text{dB}}$ | $L\downarrow$ 30% | $C_a\downarrow$ 30% | $R_a\downarrow$ 30% | $R_m\downarrow$ 30% | $R_p\downarrow$ 30% | $C_p\downarrow$ 30% |
|---|---|---|---|---|---|---|---|
| | | **$f_{-3\,\text{dB}}$ (GHz)** | | | | | |
| 5 | 18.63 | 21.3 | 18.74 | 18.63 | 18.68 | 19.39 | 22.20 |
| 8 | 21.23 | 24.74 | 21.37 | 21.23 | 21.26 | 21.84 | 25.15 |
| 12 | 23.3 | 27.31 | 23.44 | 23.3 | 23.33 | 23.88 | 27.53 |

Figure 7 shows the improvement of −3 dB bandwidth for a 30% decrease of single parasitic parameters, including $L$, $C_a$, $R_m$, $R_p$, and $C_p$ at 5 mA, 8 mA, and 12 mA, respectively. As seen in the figure, the reduction in $C_p$ led to a maximum improvement of $f_{-3\,\text{dB}}$, and a 30% reduction in $C_p$ led to an improvement of 20.02% at 5 mA, 18.46% at 8 mA, and 18.15% at 12 mA. At the same time, we noticed that the effect of $R_a$, $C_a$, $R_m$, and $R_p$ on the bandwidth was weak as the injection current increased. For instance, a 30% reduction of $C_a$ led to an improvement of only 0.59%, 0.67%, and 0.60% at 5 mA, 8 mA, and 12 mA, respectively, for $f_{-3\,\text{dB}}$, and a 30% reduction in $R_a$ had barely any improvement, probably due to the extremely low value of $R_a$. A 30% reduction in $R_p$ and $R_m$ led to an increase of approximately 0.41% and 0.27% at 5 mA, 2.87% and 0.14% at 8 mA, and 2.49% and 0.13% at 12 mA for $f_{-3\,\text{dB}}$. It is also observed that as the current increased, the effect of inductance $L$ decreased; at 5 mA, the inductance is reduced by 30%, and $f_{-3\,\text{dB}}$ is improved by 12.54%, and when the current reached 12 mA, $f_{-3\,\text{dB}}$ improved by 17.21%.

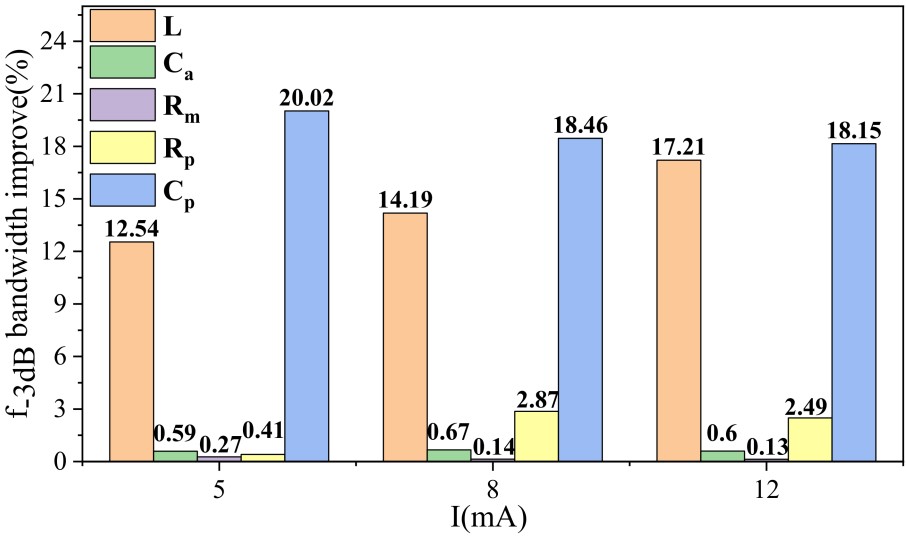

**Figure 7.** Improvement of −3 dB bandwidth for a 30% decrease of single parasitic parameters, including $L$, $C_a$, $R_m$, $R_p$, and $C_p$ at 5 mA, 8 mA, and 12 mA, respectively.

It should be noted that our results are different from those of previous works [11,14,26]. This shows that $C_a$ is not the most important factor affecting the −3 dB bandwidth because the conventional method of determining the most significant effect on the bandwidth of an element is by reducing the corresponding value by 30% and then calculating the value of the change in the $f_{-3\,dB}$ bandwidth. However, for the same component, the larger the parasitic value is the larger the 30% change in bandwidth. Our value of $C_a$ is small enough relative to the value in [11,14,26].

Consequently, the values of $R_a$, $C_a$, $R_m$, and $R_p$ based on our six-layer oxide structure are already low enough; therefore, further enhancement of the bandwidth should focus on

minimizing $C_p$ and $L$, for example, adjusting the BCB flattening height and area, replacing other flattened materials, and optimizing the pad size.

## 4. Conclusions

We proposed a modified parasitic extraction method that described the extrinsic behaviour of high-speed oxide-confined VCSELs well and easily extracted the values of all parasitic parameters, and the modified method improves the parasitic extraction model. In addition, we fabricated and characterized a six-oxide layer structure VCSEL, and a bandwidth of 23.3 GHz was achieved. The low capacitance advantage of the six-oxide layer structure was demonstrated by extracting the parasitic resistance capacitance of the device by simulation. Then, we compared and analysed the parasitic factors of the different oxide aperture diameters and mesa. The relationship between the parasitic value and the diameter of the oxidation aperture and the mesa was quantitatively analysed, and the pad capacitance is proportional to the mesa diameter and current. Finally, we investigated the extrinsic bandwidth limitations and gave improvement suggestions that decreasing pad capacitance can increase the bandwidth.

**Author Contributions:** Methodology, H.T.; validation, C.T. and H.T.; investigation, H.T. and Y.W.; data curation, S.T. and Y.W.; writing—review and editing, H.T. and C.T.; supervision, C.T. and Z.W.; project administration, C.T. and L.W. All authors have read and agreed to the published version of the manuscript.

**Funding:** This work was supported by the National Key Research and Development Program of China (2018YFB2201000) and the K.C. Wong Education Foundation (GJTD-2020-10).

**Institutional Review Board Statement:** Not applicable.

**Informed Consent Statement:** Not applicable.

**Data Availability Statement:** Not applicable.

**Conflicts of Interest:** The authors declare no conflict of interest.

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
