# Peer review of "Parasitic Parameters Extraction of High-Speed Vertical-Cavity Surface-Emitting Lasers"

_applsci, doi:10.3390/app12126035_

Round 1

Reviewer 1 Report

Summary: The authors present a study on high speed 6 oxide aperture VCSEL an apply an improved circuit model to extract external electrical parameters based on S11 measurement. The manuscript quality is good, and after some improvements I recommend publication.

More detailed comments:
Distinguish between parasitic (factor, element, effect) and parasite. Parasite reminds more of the biologigal domain.

Please justify why the basic plate capacitor is not suitable in the beginning also.

formula typo on line 111

What do you mean on line 174? That the conclusions from both the data analysis and measurement results are equal?

Table 1, lines 185-195. The variation analysis might be better displayed more visually. Now neither the bias current or the f3dB step is not equal, so seeing the real dependency of different parameters is difficult. Please consider e.g. line plot (x=current, y1=f3dB, y2=percentage improvement). 

Figure 6: do you have fit confidence bounds to show in the plots?

Conclusions: not all the extracted parameters seem to behave linearly  to 1/diameter or diameter. Can you comment why this happens? I'm looking at factors Rm and   Cp.

Author Response

Thanks for your comments. I have written the response in the attachment

Reviewer 2 Report

Reviewers comment manuscript applsci-1738532

Title: Parasitic parameters extraction of high-speed vertical-cavity surface-emitting lasers

Authors: Haixia Tong, Yanjing Wang, Sicong Tian, Cunzhu Tong, Zhipeng Wei, Lijun Wang

The paper presents parameter extraction  of the electrical parasitic elements of a 850 nm VCSELs by fitting to measured reflection measurements. The influence of both the current and the size of the VCSEL oxide aperture on the extracted parasitics and their relative impact on the modulation bandwidth was studied, with the conclusion that the pad capacitance and inductance had the largest impact for the smallest oxide aperture.

Unfortunately, I can´t recommend the paper for publication in its present state. In both the abstract, introduction and conclusions the authors claim that they propose a modified parameter extraction model. However, this is misleading since the equivalent circuit model proposed in Figure 4 is identical to that of Ref. 14 Figure 1. Furthermore, the authors use the same method to compare the relative influence of different parameters on the bandwidth (compare Table 1 with Table IV in Ref. 14). The conclusion is somewhat different, in 14 the parasitic element Ca (mainly the diffusion capacitance) limits the bandwidth of their 7µm oxide aperture, whereas in the present work, the pad capacitance Cp is the main limiting factor for the 3µm oxide aperture. However, as authors state, this is no contradiction but most likely due to the device differences.

The abstract, introduction and summary need to be rewritten to reflect that he main novelty of the paper is not the model but that the electrical parasitics have been systematically extracted from VCSELs with different oxide apertures using the equivalent circuit model in Ref. 14. In addition, I have some questions regarding the results that need to be addressed before publication.

1. There is no discussion of the uncertainty of the extracted parameters. The extraction of six parameters from the feature-less S11-measurements in Fig. 5 a. will give a large interdependence on the different parameters. The estimated accuracy and covariance of e.g. Ca and Cp would be interesting to know.

2. A detailed discussion of physical explanation of the observed current dependence and oxide aperture dependence of the parasitic elements is lacking.

3. I miss a comparison of the extracted 3dB bandwidth from S11 measurements and the measured 3dB bandwidth of S21 measurements.

4. The meaning of the sentence  “Obviously, the results of our data analysis are the same as the measurement results.” is unclear.

5. The authors write “In addition, in references 14 and 18, the values of Cp and Rp used in the device parasitic value extraction  are based on reference 23; however, the equations in that reference are derived from a flat capacitor and are not appropriate for laser devices.” Please clarify. Reference 23 has the title  “Solving least squares problems”. Why is it not applicable to laser devices?

6. The equation of Ca is misprinted.

7. Please check the language. In Conclusion it is written “which suggests that the oxidation aperture diameter is inversely proportional to the device resistance and capacitance and decreases with increasing current”

I guess it should be reversed “which suggests that the device resistance and capacitance are inversely proportional to the oxidation aperture diameter and decrease with increasing current”

Author Response

Thanks for your comments. I have written the response in the attachment,please see the attachment

Reviewer 3 Report

The manuscript by Tong et al. describes a modified parasitic extraction model to disentangle the extrinsic  behaviour of the high-speed VCSELs . The paper is very well written and experimental parameters are well supported by the experimental data. I recommend to publish this manuscript in it's present form.

Author Response

Thank you for your comments and recommendations.

Round 2

Reviewer 2 Report

I thank the authors for addressing my concerns in the response letter and made some important revisions of the paper. For example, the confidence intervals of the extracted parameters is an important improvement.

However,  the introduction is still misleading. The sentence " a first-order low-pass filter transfer function is oversimplified, and there are  large deviations between the obtained data and the actual measured data [14]" gives the impression that this applies to ref. [14].

The authors must clearly state that the equivalent circuit model used in the paper is taken from ref. 14 and ref. 21 and explain the "modification" of their method more clearly.  In the cover letter it is written "The novelty of our work is to extract the parasitic data directly by using the least squares approach based on the equivalent circuit model, instead of using the formula of Cp and Rp ." Please write then this clearly in the article and explain why  "the  equations  of  the  plate  capacitor  presented  in  the  Ref.14  and  18  are  inappropriate  for  extracting  the  capacities  and  resistance of the device." And please assess if this modification leads to significantly lower fitting error and more accurate values of the extracted parameters.
